# Fatty Acids-Based Quality Index to Differentiate Worldwide Commercial Pistachio Cultivars

**DOI:** 10.3390/molecules24010058

**Published:** 2018-12-24

**Authors:** Mahnaz Esteki, Parvin Ahmadi, Yvan Vander Heyden, Jesus Simal-Gandara

**Affiliations:** 1Department of Chemistry, University of Zanjan, Zanjan 45195-313, Iran; parvin.ahmadi@znu.ac.ir; 2Department of Analytical Chemistry Applied Chemometrics and Molecular Modelling, Center for Pharmaceutical Research (CePhaR), Vrije Universiteit Brussel (VUB), Laarbeeklaan 103, B-1090 Brussels, Belgium; yvanvdh@vub.ac.be; 3Nutrition and Bromatology Group, Department of Analytical and Food Chemistry, Faculty of Food Science and Technology, University of Vigo, Ourense Campus, E-32004 Ourense, Spain

**Keywords:** authenticity, chromatographic fingerprint, fatty acids, classification, linear discriminant analysis

## Abstract

The fatty acid profiles of five main commercial pistachio cultivars, including Ahmad-Aghaei, Akbari, Chrok, Kalle-Ghouchi, and Ohadi, were determined by gas chromatography: palmitic (C16:0), palmitoleic (C16:1), stearic (C18:0), oleic (C18:1), linoleic (C18:2), linolenic (C18:3), arachidic (C20:0), and gondoic (C20:1) acid. Based on the oleic to linoleic acid (O/L) ratio, a quality index was determined for these five cultivars: Ohadi (2.40) < Ahmad-Aghaei (2.60) < Kale-Ghouchi (2.94) < Chrok (3.05) < Akbari (3.66). Principal component analysis (PCA) of the fatty acid data yielded three significant PCs, which together account for 80.0% of the total variance in the dataset. A linear discriminant analysis (LDA) model that was evaluated with cross-validation correctly classified almost all of the samples: the average percent accuracy for the prediction set was 98.0%. The high predictive power for the prediction set shows the ability to indicate the cultivar of an unknown sample based on its fatty acid chromatographic fingerprint.

## 1. Introduction

Nuts are nutrient-dense foods, which can be part of a healthy diet. The potential of nuts consumption in the secondary prevention of heart diseases has been related to unsaturated fatty acids, vitamins, mineral constituents, and secondary metabolites, such as alkaloids, flavonoids, tannins, and anthraquinones [1]. Most epidemiologic studies have found that diets containing a high monounsaturated to saturated fat ratio as well as a polyunsaturated to saturated fat ratio may reduce serum cholesterol levels and consequently the risk of coronary artery disease [2]. The pistachio (*Pistacia vera* L.) is a nut with peculiar organoleptic characteristics. It is widely consumed as a fresh product, snack food, or ingredient of confectionery and some sausages. The pistachio consists of a shell—that is, a hard layer—surrounding the edible kernel of the nut, which also has a papery coat (skin). Oleic acid followed by linoleic and palmitic acid are the dominant components of glycerides in nuts [3]. In comparison to other edible nuts, pistachio has a higher content of monounsaturated fatty acids (MUFA) and a lower ratio of polyunsaturated to saturated fatty acids, which indicates the cholesterol-reducing potential, as well as a lower glycemic index (GI), which reduces the risk of diabetes [4,5,6,7].

Since the pistachio is a dry-climate tree, most pistachio production comes from countries with a warm, arid climate. Iran, the United States (USA), Turkey, Syria, Italy, Tunisia, and Greece are the world’s main producers of pistachio [8,9,10]. Pistachio is the most important commercial agricultural product cultivated in Iran’s tropic [11]. The export of pistachio provides the highest financial income among Iran’s non-petroleum export products. This makes Iran one of the world’s largest pistachio exporters [12]. Over the past 50 years, Iran’s pistachio production has grown dramatically, with currently around 380,000 hectares under cultivation, which produces 35,000 tons of pistachio annually [12]. The number of pistachio trees in Iran is estimated at more than 80 million. They include 17 major cultivars. Kerman province, producing 90% of pistachio in Iran, with 270,000 hectares, is the most important region for growing pistachio in Iran and the world [13]. Although Iran is one of the main centers for pistachio diversity in the world, Iran’s pistachio industry relies mainly on a limited number of cultivars, including Ohadi, Ahmad-Aghaei, Fandoghi, Akbari, and Kalle-Ghouchi, which cover more than 95% of the cultivated area [14].

Several methods have been used to distinguish between food varieties [15,16,17,18,19,20,21]. In this context, some studies have been published on the possibility of the geographical differentiation of samples from different producing countries using various chemical indicators and analytical techniques. For example, Dyszel and Pettit [22] used the triacylglycerol profile determined by HPLC and the Differential Scanning Calorimetry (DSC) peak areas to distinguish the Californian pistachio from Iranian and Turkish ones. Furthermore, Anderson and Smith [23] suggested the application of stable isotope analysis to discriminate pistachio samples from the USA, Asia, and Mediterranean countries. In many studies of different groups, it is assumed that the variations in the fatty acid composition of pistachio nuts could be attributed to the different geographical origins of the product [3,24].

After harvesting pistachios, in the sales and marketing sector, the homogeneity (being the same cultivar) of the product is of great importance. Such a product can be sold at a more satisfactory price. Therefore, pistachio sorting is very important for increasing the product value and ease of processing, for example, separating closed-shell from open-shell pistachios, and at a wider level, separating different pistachio varieties from each other. Kouchakzadeh and Brati [25] investigated the different physical properties of pistachio samples from five varieties, i.e. Ahmad-Aghayee, Akbari, Badami, Fandoghi, and Kalle-Ghouchi, for classification purposes according to variety. Omid et al. [26] developed an automatic trainable classifier, which was based on a combination of acoustic signals and an artificial neural network (ANN), for separating four varieties of pistachio nuts: Kalle-Ghouchi, Akbari, Badami, and Ahmad-Aghaei.

Pattern recognition and classification methods, which are widely used in various fields, have been given special attention in food chemistry and process monitoring in recent years. Classification is one of the main topics of chemometrics, and tries to develop a mathematical model for identifying the appropriate class of each object using experimental measurements [27]. The constructed mathematical model then is used to recognize the class membership of new objects of an unknown source. The mathematical model usually establishes a relationship between a set of variables that have been derived from chemical and analytical measurements, and one or a set of qualitative variables (e.g. variety) in order to separate the samples of each class. Since these models are constructed based on defined classes and the relationships among these classes, they are also known as supervised pattern recognition methods. In contrast, in unsupervised pattern recognition methods, there is no prior knowledge about the membership of the objects that are used, and the major concern is to reveal the organization of patterns in order to discover similarities and differences between the objects. Among the pattern recognition techniques, principal component analysis (PCA) is most frequently used as an unsupervised technique for exploratory data analysis, while linear discriminant analysis (LDA) methods are widely used as supervised pattern recognition methods [28,29,30].

The chromatographic fingerprint is a comprehensive approach to discriminate complex samples and assess their authenticity. In this approach, the entire chromatographic profile is applied by chemometrics techniques to rationalize complex chemical profiles and identify specific patterns of samples. Due to the complexity of the chromatographic profiles, it is practically impossible to perform quantitative and qualitative analyses of all the individual components. The main innovation of the fingerprint approach in multivariate classification is that it does not require the identification of the individual components in a chromatographic profile. Peak areas or the entire profile can be used without the identification of the corresponding substances [31,32]. Nowadays, modern separation devices, such as high-performance liquid chromatography (HPLC), gas chromatography (GC), and capillary electrophoresis (CE) are able to produce fingerprints of compounds in highly complex samples [33]. These chromatographic fingerprints have already been recognized as powerful tools for identifying, classifying, and recognizing samples. This approach has been used in several application fields, such as the classification, detection of adulteration [15,16,17,18,19,20,21,34], food authentication, and the identification of the geographical origin of the food products [15,35].

The sensory stability of pistachio nut varieties during storage using descriptive analysis combined with chemometrics can help producers in the management of storage length, and more particularly, export circumstances [36]. In another work, the chemometric data of fatty acids and crude fat are used to characterize the varieties of coffee [37]. In this sense, this paper reports on the employment of gas chromatographic fatty acid fingerprints in combination with multivariate data analysis to classify pistachio from different cultivars in Iran. Our findings are on the differences in fatty acid composition amongst the studied cultivars, together with the proposal of a quality index based on the oleic/linoleic acid (O/L) ratio, since oleic acid is monounsaturated, and its higher levels contribute to a higher oxidative stability and a large shelf life.

## 2. Materials and Methods

### 2.1. Chemicals and Reagents

The sources of chemicals and reference materials were as follows: methanol, sulfuric acid (98%), and sodium hydroxide (NaOH) were from Merck (Darmstadt, Germany); light petroleum ether (bp 40–60 °C, analytical grade) was from Daejung (Shiheung, Gyeonggi-Do, Korea).

### 2.2. Sample Collection

Pistachio samples were obtained from the Pistachio Research Center in Kerman, Iran. Five pistachio cultivars, which are commercially important worldwide and widely grown in Iran, were selected for this research. The cultivars are Akbari, Ahmada-Ghaie, Kalle-Ghouchi, Chrok. and Ohadi. The samples were obtained from Rafsanjan, which is located in the Kerman province, during the harvesting period of 2016. Two kg of pistachio was selected from each cultivar. The further sampling was followed a random selection of 30 pistachios from each cultivar. The pistachios were oven-dried at 30 °C for at least five days, and then stored in a refrigerator at 4 °C until further analysis.

### 2.3. Preparation of Pistachio Samples

The sample preparation process includes two steps: crude oil extraction and methyl esterification.

Fatty acids were transformed into their methyl esters using methanolic sodium hydroxide solution in order to be analyzed by gas chromatography. The details of the process have been described elsewhere [16].

### 2.4. Chromatographic Conditions

A gas chromatograph (7890N series, Agilent Technologies, Santa Clara, CA, USA) with a flame ionization detector (FID) and split/splitless injector carried out the analysis of the fatty acid methyl esters (FAMEs). Separation was performed based on a DB-WAX fused silica capillary (30 m × 0.25 mm, 0.25-μm film thickness; ARudent J and W Scientific, Folsom, CA, USA). Hydrogen gas for FID was generated with a hydrogen generator (OPGU-1500S, Shimadzu, Kyoto, Japan) at a flow rate of 30 mL min^−1^, the flow rate of air for FID was 350 mL min^−1^, and the carrier gas was nitrogen, with a flow rate of 1.0 mL min^−1^. The FID and injector temperatures were set at 220 °C and 250 °C, respectively. A volume of 1.0 µL of FAMEs, dissolved in petroleum ether, was injected directly into the gas chromatograph for analysis using a split ratio of 30:1. The “hot needle injection” technique was used in order to improve the repeatability. Oven temperature was maintained at 50 °C for one minute, and then programmed to 200 °C at a rate of 25 °C/min and then further increased at three °C/min to reach 230 °C, which was maintained for 13 min. Thus, the total time of one GC run was about 30 min. The peak areas of the FAMEs were determined by the ChemStation software, and then used for multivariate data analysis.

### 2.5. Multivariate Data Analysis

The dataset of the chromatograms of pistachio nuts was split into two: a calibration (100 samples) and a test set (50 samples). The samples from each cultivar, comprising 30 samples, were randomly divided into two categories: 20 samples for the calibration and 10 samples for the prediction set. The calibration set was used for unsupervised pattern recognition (PCA) and the development of the calibration models of LDA, while the test set then was used to establish the sensitivity and specificity of the models. In order to evaluate the effect of the different fatty acids, before applying data analysis, the peak areas were auto-scaled by subtracting sample means and then dividing the resulting difference by the corresponding standard deviations. PCA and LDA were implemented with an in-house program that was written in Matlab (version 6.5; Mathworks, Natick, MA, USA).

#### 2.5.1. Principal Component Analysis (PCA)

PCA is a commonly applied linear transformation method that is used for dimension reduction, visualization, and the exploration of multivariate data. PCA is a systematic method for analyzing multivariate data by producing new orthogonal variables, which are called principal components, and are obtained as linear combinations of the original variables. PCA is usually able to describe most of the variation in the data in the first few components. This method is also an unsupervised classification technique, projecting multidimensional data into lower dimensions with a minimal loss of information. It is employed for understanding data patterns and anomaly detection.

#### 2.5.2. Linear Discriminant Analysis (LDA)

LDA is a well-known supervised pattern recognition method that uses linear combinations of original variables to build a classifier model. The main aim of LDA is to find vectors by which the projection of points from the original space lead to maximum separation between the classes. The eigenvectors are obtained by maximizing the ratio of the between-group variance to the within-group variance. Simultaneously, the LDA method can be used for feature extraction, dimension reduction, and discrimination purposes.

## 3. Results and Discussion

Figure 1 shows a typical GC chromatogram of FAMEs obtained from an Iranian pistachio sample. Gas chromatography mass spectrometry (GC-MS) was carried out to identify the fatty acid composition of pistachio kernel oil. Palmitic acid (C16:0) as saturated fatty acid, and oleic acid (C18:1) and linoleic acid (C18:2) as unsaturated fatty acids, are the major compounds in the chromatogram. Other types of fatty acids, including palmitoleic acid (C16:1), stearic acid (C18:0), linolenic acid (C18:3), arachidic acid (C20:0), and gondoic acid (C20:1) are found as minor compounds. Myristic acid (C14:0), Margaric acid (C17:0), and heptadecenoic acid (C17:1) were present in all of the samples in trace amounts, and they were not considered for further statistical analysis. According to Satil et al. [38], these minor acids do not exceed 0.5%. Table 1 shows the fatty acid composition of the 150 samples. The principal fatty acids were palmitic (8.35–10.53%), palmitoleic (0.45–0.88%), stearic (0.98–1.8%), oleic (60.71–69.84%), linoleic (18.37–27.39%), linolenic (0.26–0.39%), arachidic (0.12–0.19%), and gondoic acid (0.41–0.59%). The fatty acid percentages that were obtained in this work were close to other reports about Iranian pistachio [3,39].

Table 2 shows the range and the mean of fatty acids for the five pistachio cultivars. The results show that the sum of oleic and linoleic acids accounts for almost 80% of the total fatty acids detected in pistachio samples, which is also seen for other nuts, such as peanut [40]. As it can be seen in Table 2, in the varieties of Akbari, Kalle-Ghouchi, and Ohadi, the oleic acid content is Ohadi < Kalle-Ghouchi < Akbari, while for linoleic acid, this trend was Akbari < Kalle-Ghouchi < Ohadi. Our results are in agreement with those obtained by Roozban et al. [41], which determined oleic and linoleic acid in Akbari, Kaleghuchi, and Ohadi varieties. The same trend in oleic acid and linoleic acid content were reported as in our results.

It is well-known that cultivar and environmental factors affect the composition, and consequently the price, of food from plants. For nuts, oil is one of the main derived products, and therefore, its quality and characteristics of its fatty acid profile are very important. The storage quality of nuts depends on the relative ratio of their saturated and unsaturated fatty acids [42]. The oxidative rancidity of most nut oils increases with increasing levels of polyunsaturated fatty acids. Therefore, the higher the unsaturation, the lower the quality of the oil. The ratio of oleic to linoleic acid (O/L), which is called the “quality index”, is commonly used as a measure to predict the shelf life and stability of the oil. A higher O/L value represents greater chemical stability and longer shelf life. The quality index of the cultivars that are assayed in this work is reported in Table 2. The highest quality index of 3.66 corresponds to Akbari, and the lowest of 2.35 was recorded for Ohadi.

In the next step, the areas of the specified peaks in the chromatograms were used as input parameters for fingerprint analysis. The resulting two-dimensional matrix (150 samples and eight peak areas, 75 × 8) was used for subsequent supervised and unsupervised pattern recognition analysis. Auto-scaling was used as data pretreatment for all applied methods.

### 3.1. Unsupervised Pattern Recognition Analysis Using PCA

Principal component analysis was applied to the data matrix of the eight major fatty acids in five pistachio cultivars. PCA was performed on the standardized peak areas matrix to generate PCs comprising a new set eight orthogonal variables. The analysis showed that 47.2% of the total variation was explained by the first principal component, while 64.9% was explained by the first two, and 80.0% was explained by the first three. Figure 2a shows the scores of the first three principal components, illustrating the distribution of the different pistachio cultivars. As it can be seen, the different cultivars are discriminated rather clearly. These preliminary results demonstrate the feasibility of discriminating the Iranian pistachio varieties based on their fatty acid compositions.

In order to evaluate the relationship between parameters, the PC1–PC2 loading plot (Figure 2b) was also examined. A loading plot visualizes the weight of the variables on a given score. Variables that are far from the origin have the highest weight (loading). The correlation between two variables can be regarded simply as the cosine of the angle enclosed between their vectors. Positive correlations correspond to angles below 90 degrees, and negative correlations correspond to angles above 90 degrees, while uncorrelated variables show right angles (angles equal to 90 degrees).

In Figure 2b, the PC1–PC2 loading plot shows positive PC1 loadings of C16:0, C16:1, and C18:2, and negative loadings of the other five fatty acids. The loading plot revealed negative correlations between C18:1 and C18:2, on the one hand, and between C16:1 or C16:0 and C20:1 on the other (the correlation was close to −1 based on their cosine angle, 180°). The correlation between C20:0 and C18:0 is close to +1 based on their cosine angle, ~0°. Table 3a, which shows the mutual Pearson correlations of the fatty acids, confirms these results. As observed, the correlation between C18:0 and C20:0 is equal to 0.905, while that between C18:1 and C18:2 is equal to −0.988, and that between C16:0 and C20:0 is equal to −0.921. Moreover, the mutual correlations of C16:0 and C18:1, C16:0 and C18:2, C16:1 and C18:3, C18:1 and C20:1, and C18:3 and C20:1 is close to zero, because their angles are close to 90°. This is also shown in Table 3a, in which the mutual correlation between the above fatty acids pairs are −0.039, −0.109, −0.064, −0.112, and 0.112, respectively. The angles between parameters C16:0 and C16:1, and between C18:1 and C18:3 illustrate also that these are quite highly correlated.

### 3.2. Supervised Pattern Recognition Analysis Using LDA

Linear discriminant analysis was applied to the above-mentioned peak areas in the chromatograms of the pistachio samples from five cultivars, in order to develop a mathematical model for their classification and identification. The LDA model was created using the training set consisting of 100 samples, while 50 samples were used as the test set to validate the predictive properties of the model. The calibration data matrix was obtained by recording the chromatograms of 100 oils (pistachio samples) and eight variables (peak areas in each chromatogram). The pistachio cultivars are the categorized dependent variables, while the independent variables include the peak areas of the fatty acids on the chromatograms.

A probability density function of the projected points [15,43], using the first, second, and the third projection vector, is shown in Figure 3A. As it is seen in Figure 3A (a), Akbari, Kalle-Ghouchi and Ohadi are separated based on the first LDA eigenvector. Figure 3A (b) illustrates that again Akbari, Kalle-Ghouchi, and Ohadi are discriminated based on the second eigenvector. Figure 3A (c) also shows that Ahmad-Aghaei and Chrok are separated. Therefore, it seems that the discrimination of cultivars would be feasible using three first eigenvectors. With this model, an acceptable distinction between all of the classes was obtained (Figure 3B).

The cultivars of the calibration samples were found correctly for almost all of the samples based on leave-one-out cross-validation (99.0%). Furthermore, bootstrapping as a resampling method was also used to generate different distributions of predicted classification results and subsequently assess classification accuracies. Bootstrapping was performed by randomly selecting samples for individual subpopulations 50 times, including five samples for prediction in each step; the average accuracy of the corresponding method was 98.75%. The developed LDA model was subsequently applied to the data of the 50 test set samples in order to predict their cultivars. The detailed LDA classification results are listed in Table 3B. The classification accuracy, sensitivity, and specificity values were reported in order to evaluate the classification performance of the model. The classification sensitivity for each class, which was calculated as TP/(TP+FN) (TP: True Positive, FN: False Negative), and the specificity, which was calculated as TN/(TN + FP) (TN: True Negative, FP: False Positive) are tabulated in Table 3B. The model provided an average accuracy of 98.0%, while the average sensitivity and specificity were 98.0% and 99.6%, respectively. The high percentage of classification accuracy of the training data indicates a reasonable relationship between the fatty acid chromatographic data and the cultivars, while the high correct accuracy percentage for the prediction set demonstrates the ability of the developed model to indicate the cultivar of an unknown pistachio sample based on the fatty acid profile.

## 4. Conclusions

In this study, the potential of GC coupled to supervised and unsupervised pattern recognition methods for the traceability and classification of Iranian pistachio nuts based on their fatty acid composition was demonstrated. A classification model with high sensitivity was constructed to predict the cultivars of the samples, which were then evaluated by cross-validation and additionally by the external test set. To the best of our knowledge, the classification of these cultivars has not yet been reported yet. The outcome for this classification approach is very satisfying: the cultivars of the validation samples were classified correctly by LDA modeling in more than 98% of cases. The models that were built were very sensitive and highly specific for authenticating the provenance of pistachio nuts.

## Figures and Tables

**Figure 1 molecules-24-00058-f001:**
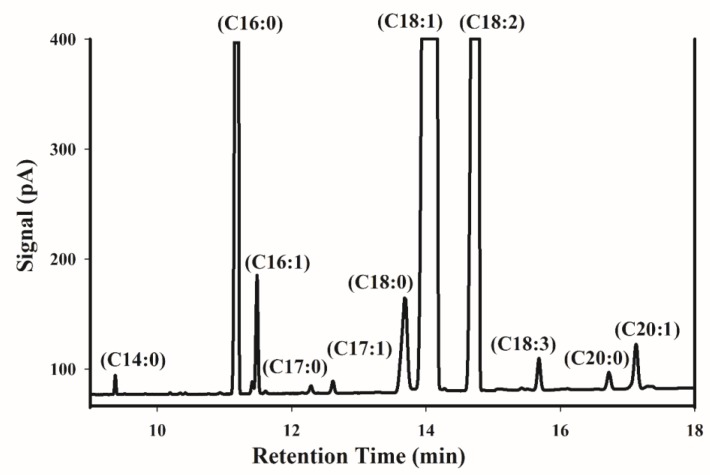
Representative fatty acid methyl esters gas chromatography (GC) chromatogram of an Iranian pistachio sample.

**Figure 2 molecules-24-00058-f002:**
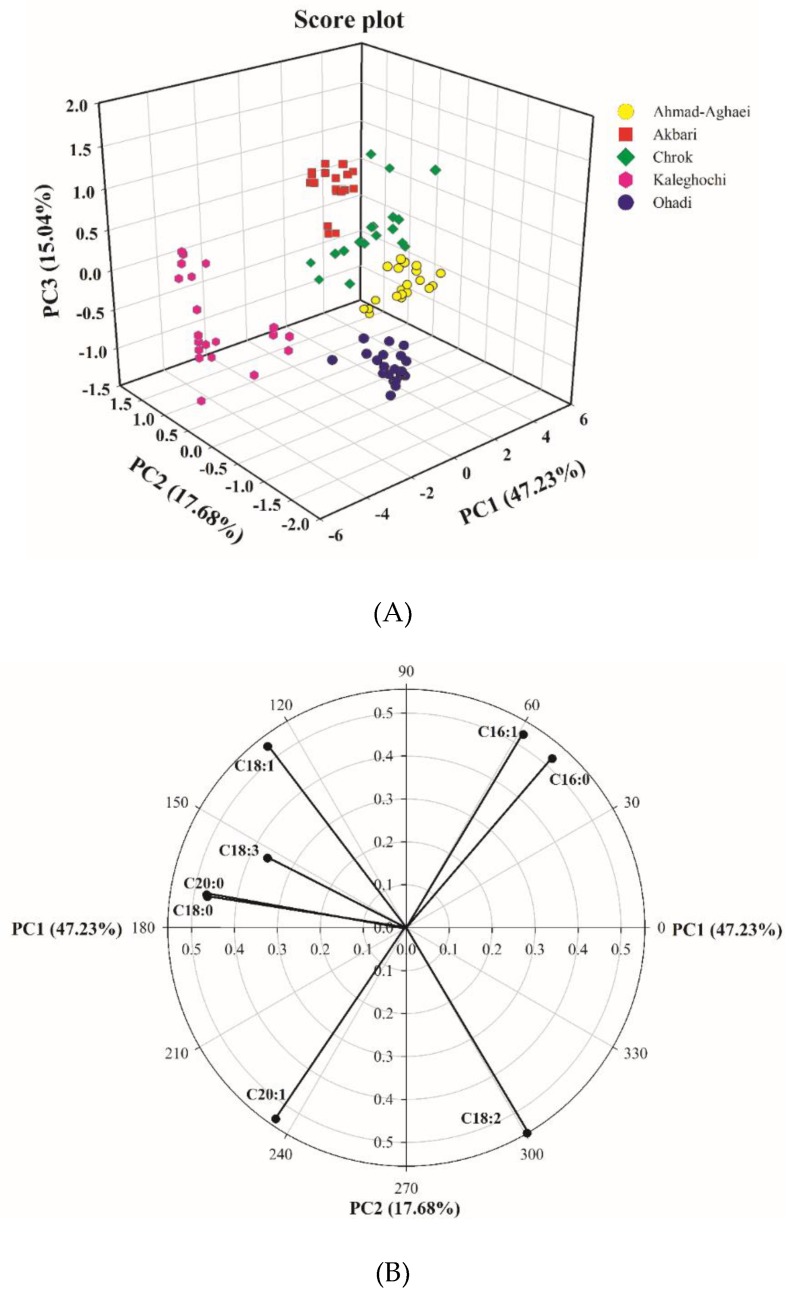
PC1–PC2–PC3 score plot from the matrix of the gas chromatographic peak areas of the fatty acid fingerprints from the pistachio samples (**A**), and PC1–PC2 loading plot from the matrix of the gas chromatographic peak areas’ fatty acid fingerprints from the pistachio samples (**B**).

**Figure 3 molecules-24-00058-f003:**
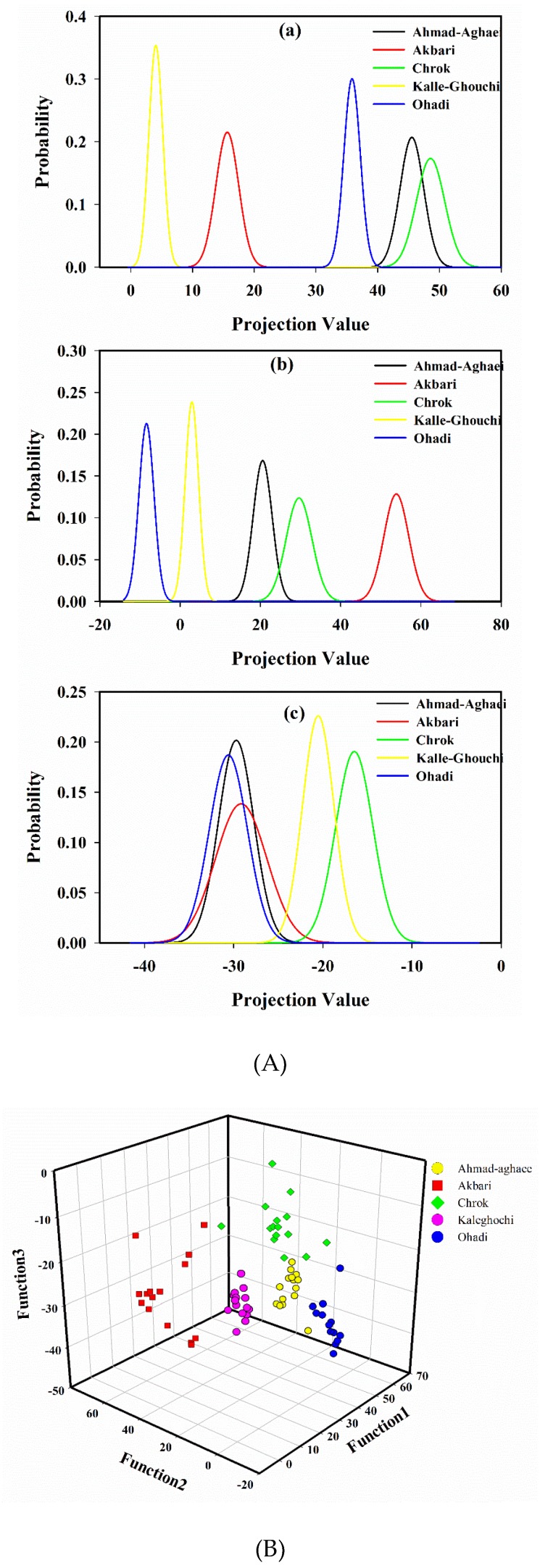
The probability density function of projected points of the peak areas from the fatty acid fingerprints by linear discriminant analysis using the (**a**) first, (**b**) second, and (**c**) third projection vector (**A**), and scatter plot of the calibration samples in 3D space defined by the three discriminant functions of the LDA model, which was constructed to classify pistachio samples according to their cultivar (**B**).

**Table 1 molecules-24-00058-t001:** Fatty acid composition (%) of pistachio oils of different cultivars.

Number of Sample	Pistachio Cultivar	Fatty Acid Composition (%)							
		C16:0 (Palmitic)	C16:1 (Palmitoleic)	C18:0 (Stearic)	C18:1 (Oleic)	C18:2 (Linoleic)	C18:3 (Linolenic)	C20:0 (Arachidic)	C20:1 (Gondoic)
1	Ahm	8.89	0.53	1.52	64.59	23.50	0.29	0.16	0.52
2	Ahm	8.87	0.52	1.51	64.51	23.61	0.28	0.17	0.53
3	Ahm	8.91	0.55	1.45	63.55	24.56	0.29	0.15	0.54
4	Ahm	8.82	0.52	1.52	64.55	23.6	0.29	0.17	0.53
5	Ahm	8.61	0.54	1.47	62.73	25.66	0.30	0.16	0.53
6	Ahm	8.90	0.53	1.53	64.43	23.64	0.29	0.16	0.52
7	Ahm	8.65	0.55	1.39	62.02	26.43	0.28	0.14	0.54
8	Ahm	8.67	0.58	1.32	61.40	27.03	0.30	0.15	0.55
9	Ahm	8.75	0.53	1.33	64.04	24.33	0.31	0.16	0.55
10	Ahm	8.78	0.58	1.26	61.33	27.07	0.30	0.14	0.54
11	Ahm	8.81	0.52	1.53	64.54	23.61	0.28	0.17	0.54
12	Ahm	8.66	0.54	1.50	63.46	24.85	0.29	0.16	0.54
13	Ahm	8.89	0.52	1.55	64.57	23.50	0.28	0.16	0.53
14	Ahm	8.77	0.53	1.43	62.28	26.03	0.28	0.16	0.52
15	Ahm	8.91	0.52	1.52	64.59	23.48	0.28	0.17	0.53
16	Ahm	8.61	0.53	1.44	63.11	25.31	0.29	0.16	0.55
17	Ahm	8.89	0.52	1.53	64.62	23.46	0.28	0.17	0.53
18	Ahm	8.66	0.54	1.51	64.15	24.15	0.29	0.16	0.54
19	Ahm	8.88	0.57	1.35	61.29	26.89	0.30	0.15	0.57
20	Ahm	8.79	0.53	1.38	63.09	25.19	0.31	0.15	0.56
21	Ahm	8.82	0.59	1.42	65.10	23.08	0.32	0.16	0.51
22	Ahm	8.69	0.58	1.39	64.12	24.20	0.31	0.15	0.56
23	Ahm	8.86	0.59	1.35	65.08	23.13	0.32	0.16	0.51
24	Ahm	8.72	0.57	1.42	65.14	23.18	0.29	0.16	0.52
25	Ahm	8.86	0.59	1.37	65.09	23.10	0.32	0.16	0.51
26	Ahm	8.73	0.58	1.33	64.26	24.14	0.28	0.15	0.53
27	Ahm	8.88	0.59	1.38	65.02	23.14	0.32	0.16	0.51
28	Ahm	8.63	0.56	1.38	64.28	24.16	0.28	0.15	0.56
29	Ahm	8.59	0.54	1.46	63.03	25.41	0.28	0.16	0.53
30	Ahm	8.65	0.54	1.29	61.64	26.88	0.29	0.17	0.54
31	Akb	8.37	0.58	1.51	69.63	18.91	0.29	0.16	0.55
32	Akb	8.56	0.57	1.45	69.45	18.95	0.31	0.17	0.54
33	Akb	8.38	0.57	1.51	69.67	18.86	0.30	0.16	0.55
34	Akb	8.52	0.59	1.53	68.99	19.38	0.28	0.17	0.54
35	Akb	8.38	0.57	1.52	69.65	18.86	0.30	0.16	0.56
36	Akb	8.43	0.58	1.48	69.55	18.95	0.29	0.18	0.54
37	Akb	8.40	0.57	1.52	69.62	18.88	0.30	0.16	0.55
38	Akb	8.59	0.58	1.49	69.77	18.57	0.30	0.16	0.54
39	Akb	8.35	0.56	1.52	69.70	18.86	0.29	0.16	0.56
40	Akb	8.51	0.56	1.50	68.95	19.49	0.28	0.17	0.54
41	Akb	8.75	0.59	1.44	68.57	19.69	0.28	0.16	0.52
42	Akb	8.61	0.58	1.52	69.15	19.14	0.29	0.17	0.54
43	Akb	8.74	0.59	1.45	68.59	19.67	0.28	0.16	0.52
44	Akb	8.42	0.58	1.53	69.66	18.82	0.30	0.16	0.53
45	Akb	8.67	0.59	1.44	68.68	19.66	0.28	0.16	0.52
46	Akb	8.55	0.57	1.49	69.26	19.11	0.29	0.18	0.55
47	Akb	8.33	0.56	1.54	69.82	18.72	0.29	0.17	0.57
48	Akb	8.45	0.59	1.51	68.95	19.50	0.28	0.16	0.56
49	Akb	8.40	0.57	1.53	69.71	18.79	0.29	0.16	0.55
50	Akb	8.99	0.58	1.54	69.61	18.29	0.28	0.17	0.54
51	Akb	9.08	0.60	1.52	68.71	19.05	0.31	0.19	0.54
52	Akb	8.51	0.57	1.53	68.88	19.48	0.30	0.16	0.57
53	Akb	8.70	0.58	1.49	69.84	18.37	0.30	0.18	0.54
54	Akb	8.66	0.57	1.50	69.25	19.03	0.30	0.16	0.53
55	Akb	8.96	0.60	1.57	69.23	18.63	0.30	0.18	0.53
56	Akb	8.48	0.59	1.54	69.62	18.73	0.29	0.17	0.58
57	Akb	8.90	0.60	1.56	69.32	18.61	0.30	0.18	0.53
58	Akb	8.52	0.58	1.55	69.87	18.46	0.29	0.16	0.57
59	Akb	8.41	0.56	1.56	69.42	18.98	0.30	0.18	0.59
60	Akb	8.55	0.58	1.56	68.85	19.41	0.31	0.16	0.58
61	Chr	9.40	0.64	1.36	64.39	23.26	0.31	0.15	0.48
62	Chr	9.46	0.57	1.55	65.16	22.35	0.28	0.15	0.48
63	Chr	9.63	0.66	1.31	66.22	23.25	0.31	0.15	0.47
64	Chr	9.23	0.56	1.54	65.33	22.41	0.29	0.16	0.48
65	Chr	9.16	0.54	1.50	66.84	21.00	0.28	0.17	0.51
66	Chr	9.15	0.59	1.44	66.66	21.23	0.29	0.15	0.49
67	Chr	9.21	0.55	1.53	66.75	21.02	0.28	0.16	0.50
68	Chr	9.13	0.53	1.52	65.70	22.21	0.28	0.15	0.48
69	Chr	9.19	0.55	1.56	66.78	20.97	0.28	0.16	0.51
70	Chr	9.26	0.54	1.49	66.59	21.13	0.30	0.17	0.52
71	Chr	9.18	0.54	1.47	66.85	21.02	0.28	0.16	0.50
72	Chr	9.46	0.53	1.56	65.08	22.41	0.29	0.18	0.49
73	Chr	9.06	0.54	1.75	64.98	22.66	0.30	0.17	0.54
74	Chr	9.08	0.54	1.56	64.78	23.06	0.30	0.15	0.53
75	Chr	9.11	0.54	1.63	64.99	22.73	0.30	0.17	0.53
76	Chr	9.51	0.57	1.50	66.03	21.42	0.30	0.15	0.52
77	Chr	9.23	0.55	1.49	66.68	21.09	0.29	0.16	0.51
78	Chr	9.13	0.59	1.62	67.37	20.35	0.28	0.17	0.49
79	Chr	9.20	0.55	1.50	66.70	21.09	0.29	0.16	0.51
80	Chr	9.51	0.61	1.49	66.67	20.75	0.28	0.15	0.54
81	Chr	9.27	0.56	1.44	66.62	21.18	0.29	0.15	0.49
82	Chr	9.37	0.60	1.54	67.04	20.54	0.28	0.15	0.48
83	Chr	9.19	0.56	1.69	67.45	20.10	0.31	0.19	0.51
84	Chr	9.16	0.51	1.65	66.21	21.48	0.29	0.18	0.52
85	Chr	9.21	0.57	1.80	67.34	20.08	0.31	0.19	0.50
86	Chr	8.67	0.51	1.61	65.93	22.32	0.29	0.15	0.52
87	Chr	8.43	0.50	1.78	65.55	22.66	0.30	0.19	0.59
88	Chr	9.36	0.52	1.59	65.54	22.05	0.28	0.15	0.51
89	Chr	9.03	0.53	1.56	65.21	22.67	0.30	0.17	0.53
90	Chr	9.46	0.58	1.60	66.13	21.32	0.28	0.16	0.51
91	Kal	10.10	0.79	1.37	66.90	19.96	0.28	0.16	0.44
92	Kal	9.33	0.75	1.28	66.78	21.03	0.27	0.13	0.43
93	Kal	10.12	0.79	1.36	66.96	19.93	0.27	0.15	0.42
94	Kal	9.66	0.74	1.30	66.28	21.17	0.28	0.12	0.45
95	Kal	10.53	0.88	1.32	66.61	19.82	0.29	0.14	0.41
96	Kal	9.67	0.76	1.43	64.84	22.45	0.27	0.12	0.46
97	Kal	9.78	0.77	1.45	67.11	19.99	0.29	0.16	0.45
98	Kal	10.02	0.79	1.40	63.75	23.20	0.27	0.15	0.42
99	Kal	10.09	0.78	1.41	67.40	19.47	0.26	0.16	0.43
100	Kal	9.65	0.69	1.35	67.19	20.33	0.26	0.12	0.41
101	Kal	10.09	0.78	1.42	67.51	19.35	0.26	0.16	0.43
102	Kal	9.32	0.75	1.36	68.57	19.15	0.27	0.12	0.46
103	Kal	10.15	0.77	1.39	67.43	19.43	0.26	0.15	0.42
104	Kal	9.15	0.68	1.30	62.17	25.75	0.27	0.15	0.53
105	Kal	9.19	0.73	1.21	64.29	23.68	0.27	0.16	0.47
106	Kal	9.05	0.67	1.31	62.85	25.18	0.26	0.15	0.53
107	Kal	9.66	0.71	1.21	64.94	22.58	0.26	0.14	0.50
108	Kal	9.14	0.68	1.31	62.09	25.83	0.26	0.15	0.54
109	Kal	10.15	0.69	1.32	64.33	22.64	0.27	0.14	0.46
110	Kal	9.55	0.74	1.00	65.89	21.94	0.28	0.13	0.47
111	Kal	10.23	0.81	1.12	64.96	22.02	0.26	0.14	0.46
112	Kal	9.57	0.74	0.99	65.91	21.91	0.28	0.13	0.47
113	Kal	10.10	0.79	1.43	65.45	21.37	0.27	0.14	0.45
114	Kal	9.57	0.74	0.98	65.98	21.91	0.28	0.12	0.42
115	Kal	9.76	0.80	1.42	63.84	23.26	0.27	0.15	0.50
116	Kal	9.01	0.65	1.29	62.39	25.69	0.26	0.16	0.55
117	Kal	9.33	0.75	1.15	64.82	23.03	0.26	0.14	0.52
118	Kal	9.17	0.66	1.31	62.18	25.72	0.26	0.15	0.55
119	Kal	9.26	0.71	1.09	64.68	23.32	0.27	0.16	0.51
120	Kal	9.31	0.73	0.99	64.40	23.71	0.27	0.13	0.46
121	Oha	8.99	0.54	1.25	61.42	26.87	0.26	0.15	0.52
122	Oha	9.05	0.46	1.22	63.79	24.54	0.26	0.16	0.52
123	Oha	8.96	0.54	1.29	61.42	26.85	0.26	0.15	0.53
124	Oha	9.15	0.47	1.30	61.66	26.46	0.27	0.15	0.54
125	Oha	8.92	0.53	1.27	61.52	26.82	0.26	0.15	0.53
126	Oha	9.01	0.58	1.35	63.58	24.55	0.26	0.16	0.51
127	Oha	9.36	0.45	1.21	65.28	22.77	0.27	0.15	0.51
128	Oha	9.05	0.41	1.22	63.93	24.44	0.26	0.17	0.52
129	Oha	8.89	0.52	1.25	65.61	22.79	0.26	0.15	0.53
130	Oha	9.10	0.58	1.22	61.28	26.89	0.26	0.15	0.52
131	Oha	8.86	0.53	1.41	61.48	26.77	0.26	0.15	0.54
132	Oha	8.88	0.51	1.31	61.36	27.01	0.26	0.15	0.52
133	Oha	8.87	0.53	1.32	61.56	26.77	0.26	0.15	0.54
134	Oha	9.00	0.49	1.45	62.00	26.13	0.27	0.15	0.51
135	Oha	8.95	0.53	1.23	61.49	26.87	0.26	0.15	0.52
136	Oha	8.99	0.49	1.43	64.00	24.14	0.26	0.16	0.53
137	Oha	8.94	0.53	1.40	61.41	26.79	0.26	0.15	0.52
138	Oha	8.86	0.51	1.23	61.18	27.26	0.27	0.17	0.52
139	Oha	8.94	0.53	1.26	61.51	26.83	0.26	0.15	0.52
140	Oha	9.06	0.59	1.32	61.97	26.12	0.26	0.15	0.53
141	Oha	8.97	0.53	1.30	61.41	26.86	0.26	0.15	0.52
142	Oha	9.01	0.60	1.28	62.17	26.01	0.26	0.16	0.51
143	Oha	8.96	0.53	1.24	61.46	26.88	0.26	0.15	0.52
144	Oha	8.91	0.46	1.30	61.41	26.98	0.27	0.15	0.52
145	Oha	8.96	0.53	1.32	61.37	26.88	0.26	0.15	0.53
146	Oha	9.03	0.52	1.28	62.67	25.57	0.26	0.16	0.51
147	Oha	9.14	0.54	1.26	60.72	27.39	0.26	0.15	0.54
148	Oha	8.87	0.59	1.44	63.05	25.12	0.26	0.15	0.52
149	Oha	9.22	0.60	1.49	65.33	22.43	0.26	0.17	0.50
150	Oha	9.08	0.45	1.25	62.11	26.18	0.26	0.15	0.52

Ahm: Ahmad-Aghaei, Akb: Akbari, Chr: Chrok, Kal: Kalle-Ghouchi, Oha: Ohadi.

**Table 2 molecules-24-00058-t002:** The range and average values of the fatty acid composition (%) of pistachio oils of different cultivars.

Pistachio Cultivar	Ahm		Akb		Chr		Kal		Oha	
	Range	Average	Range	Average	Range	Average	Range	Average	Range	Average
C16:0 (Palmitic)	8.58–8.91	8.77	8.35–9.08	8.57	8.43–9.63	9.23	9.01–10.53	9.66	8.86–9.36	8.99
C16:1 (Palmitoleic)	0.52–0.59	0.55	0.56–0.6	0.58	0.50–0.66	0.55	0.65–0.88	0.74	0.45–0.6	0.52
C18:0 (Stearic)	1.26–1.55	1.43	1.44–1.57	1.51	1.31–1.80	1.55	0.98–1.45	1.28	1.21–1.49	1.30
C18:1 (Oleic)	61.38–65.09	63.72	68.57–69.84	69.34	64.39–67.46	66.05	62.09–67.43	65.28	60.71–65.29	62.30
C18:2 (Linoleic)	23.08–27.07	24.54	18.37–19.68	18.99	20.08–23.26	21.65	19.47–25.83	22.16	22.43–27.39	25.9
C18:3 (Linolenic)	0.28–0.32	0.29	0.28–0.31	0.29	0.28–0.31	0.29	0.26–0.29	0.27	0.26–0.27	0.26
C20:0 (Arachidic)	0.14–0.17	0.16	0.16–0.19	0.16	0.15–0.19	0.16	0.12–0.16	0.14	0.15–0.17	0.15
C20:1 (Gondoic)	0.51–0.57	0.54	0.52–0.59	0.54	0.47–0.59	0.51	0.41–0.55	0.47	0.50–0.54	0.52
Quality Index	2.60		3.66		3.05		2.94		2.40	

Ahm: Ahmad-Aghaei, Akb: Akbari, Chr: Chrok, Kal: Kalle-Ghouchi, Oha: Ohadi.

**Table 3 molecules-24-00058-t003a:** (**a**)

Fatty Acid	C16:0	C16:1	C18:0	C18:1	C18:2	C18:3	C20:0	C20:1
C16:0	1							
C16:1	0.745	1						
C18:0	−0.312	−0.365	1					
C18:1	−0.039	0.209	0.477	1				
C18:2	−0.109	−0.315	−0.462	−0.988	1			
C18:3	−0.247	−0.064	0.419	0.46	−0.433	1		
C20:0	−0.304	−0.338	0.905	0.47	−0.454	0.406	1	
C20:1	−0.921	−0.730	0.3	−0.112	0.241	0.112	0.341	1

**Table 3 molecules-24-00058-t003b:** (**b**)

		Group Predicted								
Original Group	Number of Cases	Ahm	Akb	Chr	Kal	Oha		Accuracy (%)	Sensitivity	Specificity
Ahm	10	10	0	0	0	0		100	100	98
Akb	10	0	10	0	0	0		100	100	100
Chr	10	1	0	9	0	0		90	90	100
Kal	10	0	0	0	10	0		100	100	100
Oha	10	0	0	0	0	10		100	100	100
							Predictive ability	98.0	98.0	99.6

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
