# Peer review of "Fatty Acids-Based Quality Index to Differentiate Worldwide Commercial Pistachio Cultivars"

_molecules, 2018, doi:10.3390/molecules24010058_

Round 1

Reviewer 1 Report

The authors systematically describe the application of chemometric approaches on a classification of agricultural products. But, I don't agree with the definition of “quality index” in Line 211-217. Why is O/L really related to the degree of rancidity? Both are the unsaturated fatty acids. Authors should conservatively describe the characteristics of flavor in sensory evaluation, and appropriate citations in literatures as following to compare.

1. Mahdi Ghasemi-Varnamkhasti, “Sensory stability of pistachio nut (Pistacia vera L.) varieties during storage using descriptive analysis combined with chemometrics” Engineering in Agriculture, Environment and Food 8 (2015) 106-113.

2. Y-C Hung, P Chen and L-Y Chen, “Advanced classification of coffee beans with fatty acids profiling to block information loss” Symmetry 10 (2018) 529-539.

Author Response

Referee 1:

The authors systematically describe the application of chemometric approaches on a classification of agricultural products. But, I don't agree with the definition of “quality index” in Line 211-217. Why is O/L really related to the degree of rancidity? Both are the unsaturated fatty acids.

According to Davis et al. (2013), the oleic/linoleic acid (O/L) ratio largely influences oxidative stability and hence peanut shelf life. Traditional peanut seed have O/L ratios near 1.5–2.0; however, many new cultivars are ‘‘high oleic’’ with O/L ratios ≥9.

In this sense, since oleic acid is monoinsaturated, its higher levels contribute to a higher oxidative stability and a large shelf life.

Davis et al. (2013). Refractive Index and Density Measurements of Peanut Oil for Determining Oleic and Linoleic Acid Contents. J Am Oil Chem Soc, 90:199–206

Authors should conservatively describe the characteristics of flavor in sensory evaluation, and appropriate citations in literatures as following to compare.

1. Mahdi Ghasemi-Varnamkhasti, “Sensory stability of pistachio nut (Pistacia vera L.) varieties during storage using descriptive analysis combined with chemometrics” Engineering in Agriculture, Environment and Food 8 (2015) 106-113.

2. Y-C Hung, P Chen and L-Y Chen, “Advanced classification of coffee beans with fatty acids profiling to block information loss” Symmetry 10 (2018) 529-539.

The recomended references were used as follows in lines 124-130: < Sensory stability of pistachio nut varieties during storage using descriptive analysis combined with chemometrics can help producers for the management of storage length and more particularly export circumstances (Ghasemi-Varnamkhasti, 2015). In another work, the chemometric data of fatty acids and crude fat are used to characterize the varieties of coffee (Hung et al., 2018). In this sense, this paper reports on the employment of gas chromatographic fatty-acid fingerprints in combination with multivariate data analysis to classify pistachio from different cultivars in Iran>.

Reviewer 2 Report

This article shows fatty acid compositions in pistachio which are cultivated in different districts. Fatty acid composition in food has an effect on nutrition metabolism of living body. So analysis of fatty acid composition of food is important. In this article, gas chromatography is used for fatty acid analysis. It is reasonable. Analytical data shown in the results are reliable.

There are some obscure points as follows.

1. Aim of the study is not clear. Building of pattern recognition methods of fatty acid with gas chromatography is already done, and this article has low novelty. However, the differences in fatty acid composition between cultivars seem to be new findings.

2. What does O/L ratio decrease with linoleic acid increasing mean in nutrition metabolism?

3. What kinds of fatty acid compositions in pistachio is good? Describe authors’ idea with effect on nutritional effect for living body.

Author Response

Referee 2:

This article shows fatty acid compositions in pistachio which are cultivated in different districts. Fatty acid composition in food has an effect on nutrition metabolism of living body. So analysis of fatty acid composition of food is important. In this article, gas chromatography is used for fatty acid analysis. It is reasonable. Analytical data shown in the results are reliable.

Many thanks for your overall positive comment on the interest about our contribution to the journal.

There are some obscure points as follows.

1. Aim of the study is not clear. Building of pattern recognition methods of fatty acid with gas chromatography is already done, and this article has low novelty. However, the differences in fatty acid composition between cultivars seem to be new findings.

The novelty of the aim was clarified as follows in lines 130-133: <It is also relevant and new our findings on the differences in fatty acid composition amongst studied cultivars, together with the proposal of a quality index based on the oleic/linoleic acid (O/L) ratio, since oleic acid is monounsaturated and its higher levels contribute to a higher oxidative stability and a large shelf life>.

2. What does O/L ratio decrease with linoleic acid increasing mean in nutrition metabolism?

There are detrimental effects of elevated dietary linoleic acid on human health related to its role in inflammation (Jandacek, 2017). Anyway, we studied the oleic/linoleic acid (O/L) ratio as a pitachio nut quality index, since oleic acid is monounsaturated and its higher levels contribute to a higher oxidative stability and a large shelf life.

Jandacek, R.J. (2017). Linoleic acid: a nutritional quandary. Healthcare (Basel), 5(2), 25-32.

3. What kinds of fatty acid compositions in pistachio is good? Describe authors’ idea with effect on nutritional effect for living body.

According to Thijssen and Mensink (2005), with realistic intakes of stearic, oleic, and linoleic acids, differences between their effects on the serum lipoprotein profile are small. For oleic/linoleic ratios between 2.40 and 3.66 for the studied varieties do not any Nutritional diffrences are expected . However, for the nuts with the highest O/L ratio of 3.66 (Akbari variety), higher oxidative stability and a large shelf life is clear.

Thijssen, M.A., & Mensink, R.P. (2005). Small differences in the effects of stearic acid, oleic acid, and linoleic acid on the serum lipoprotein profile of humans. The American Journal of Clinical Nutrition, 82(3), 510–516.

Round 2

Reviewer 2 Report

Authors revised the manuscript well according to my opinions. This manuscript is improved to be acceptable for publication.